# Cancer-Associated Fibroblasts as a Common Orchestrator of Therapy Resistance in Lung and Pancreatic Cancer

**DOI:** 10.3390/cancers13050987

**Published:** 2021-02-27

**Authors:** Andreas Domen, Delphine Quatannens, Sara Zanivan, Christophe Deben, Jonas Van Audenaerde, Evelien Smits, An Wouters, Filip Lardon, Geert Roeyen, Yannick Verhoeven, Annelies Janssens, Timon Vandamme, Peter van Dam, Marc Peeters, Hans Prenen

**Affiliations:** 1Center for Oncological Research (CORE), Integrated Personalized and Precision Oncology Network (IPPON), University of Antwerp, B2610 Antwerp, Belgium; andreas.domen@uantwerpen.be (A.D.); delphine.quatannens@uantwerpen.be (D.Q.); christophe.deben@uantwerpen.be (C.D.); Jonas.vanaudenaerde@uantwerpen.be (J.V.A.); evelien.smits@uza.be (E.S.); an.wouters@uantwerpen.be (A.W.); filip.lardon@uantwerpen.be (F.L.); geert.roeyen@uza.be (G.R.); yannick.verhoeven@uantwerpen.be (Y.V.); timon.vandamme@uantwerpen.be (T.V.); peter.vandam@uza.be (P.v.D.); marc.peeters@uza.be (M.P.); 2Department of Oncology, Antwerp University Hospital (UZA), 2650 Edegem, Belgium; 3Cancer Research UK, Beatson Institute, Glasgow G611BD, UK; Sara.Zanivan@glasgow.ac.uk; 4Institute of Cancer Sciences, University of Glasgow, Glasgow G611QH, UK; 5Department of Hepatobiliary Transplantation and Endocrine Surgery, Antwerp University Hospital (UZA), 2650 Edegem, Belgium; 6Department of Pulmonology & Thoracic Oncology, Antwerp University Hospital (UZA), 2650 Edegem, Belgium; annelies.janssens@uza.be; 7Gynaecologic Oncology Unit, Antwerp University Hospital (UZA), 2650 Edegem, Belgium

**Keywords:** cancer associated fibroblasts, therapy resistance, lung cancer, pancreatic cancer

## Abstract

**Simple Summary:**

The tumor microenvironment (TME) is increasingly believed to be involved in therapy resistance and disease progression of different cancer types. Cancer-associated fibroblasts (CAFs) are prominent components and a central player in creating this TME. In particular, lung and pancreatic cancer, two solid tumor types associated with therapy resistance and poor long-term prognosis have clear evidence of CAFs. In order to provide a more holistic approach of therapy resistance, we highlight different mechanisms of CAF contribution to tumor progression and provide a comprehensive review on how CAFs affect their response to clinical therapy and prognosis, and present an overview of novel therapies and future perspectives involving CAF-targeting agents for both cancer types.

**Abstract:**

Cancer arises from mutations accruing within cancer cells, but the tumor microenvironment (TME) is believed to be a major, often neglected, factor involved in therapy resistance and disease progression. Cancer-associated fibroblasts (CAFs) are prominent and key components of the TME in most types of solid tumors. Extensive research over the past decade revealed their ability to modulate cancer metastasis, angiogenesis, tumor mechanics, immunosuppression, and drug access through synthesis and remodeling of the extracellular matrix and production of growth factors. Thus, they are considered to impede the response to current clinical cancer therapies. Therefore, targeting CAFs to counteract these protumorigenic effects, and overcome the resistance to current therapeutic options, is an appealing and emerging strategy. In this review, we discuss how CAFs affect prognosis and response to clinical therapy and provide an overview of novel therapies involving CAF-targeting agents in lung and pancreatic cancer.

## 1. Introduction

As our knowledge on cancer biology continuously evolves, the tumor microenvironment (TME) has gained interest over the past decade. Before, cancer research has been focusing primarily on the malignant cell itself [1]. However, cancers are not simply autonomous neoplastic cells [1] but interact with a complex surrounding ecosystem of stromal cells, known as the TME [2]. The TME consists of cancer-associated fibroblasts (CAFs), extracellular matrix (ECM), endothelial cells, and infiltrating innate and adaptive immune cells (e.g., natural killer (NK) cells and T cells, respectively) [3]. CAFs, the most prominent [4] and key component [5] of the TME, can either originate from normal fibroblasts, which are non-epithelial, non-endothelial, non-immune resting mesenchymal cells in diverse connective tissue components [6], or other precursor cells such as bone marrow-derived mesenchymal stem cells, epithelial cells, carcinoma cells, endothelial cells, pericytes, smooth muscle cells, adipocytes, fibrocytes, or specialized cells such as pancreatic stellate cells (PSCs) [3]. However, as there are no unique markers for fibroblasts that are not expressed by other cell types, it is hard to define a CAF [3,7]. Therefore, all fibroblasts associated with tumors [6] that are negative for epithelial, endothelial, and leukocyte markers with an elongated morphology and lacking the mutations found within cancer cells might be considered CAFs [7].

Fibroblasts reside in the ECM of healthy tissue to sustain normal tissue homeostasis and are found in an inactive quiescent state under normal physiologic conditions [8]. As a consequence of a disturbance of tissue integrity, such as tissue injury, they become activated and capable of producing transforming growth factor-β (TGF-β) and vascular endothelial growth factor A (VEGF-A) to stimulate wound healing and angiogenesis, respectively [8,9]. Besides, activated fibroblasts are also modulators of the immune response via the secretion of cytokines (e.g., TGF-β and interleukin (IL)-6 [6]) and chemokines (e.g., C-C motif chemokine ligand (CCL) 5, C-X-C motif chemokine ligand (CXCL) 10 [6]), and additionally promote immune surveillance [1,7]. When the process is complete and the activating stimulus is attenuated, the fibroblast activation is reversed to the inactive quiescent state by reprogramming or apoptosis [6]. However, diverse stimuli (reviewed in detail [7]) can cause chronic and irreversible fibroblast activation. This includes inflammatory signals (e.g., IL-1, IL-6, and tumor necrosis factor (TNF)), TGF-β, physical changes in the ECM, contact signals with cancer cells, and DNA damage [7], resulting in a gain of proliferation, secretory phenotype, migration, and ECM production and remodeling [6]. In case of (pre)malignant lesions, persistent accumulation of cancer cells represents an ongoing tissue injury. This initiates a chronic and irreversible hyperactivated fibroblast response towards the cancer cells and eventually the rise of CAFs, with enhanced proliferative properties and self-sustained activation. Additional fibroblast recruitment and proliferation is governed by the release of growth factors (e.g., TGF-β, PDGF, and fibroblast growth factor (FGF) 2) by the cancer cells and infiltrating immune cells [6]. Once CAFs are activated, they can contribute to tumor progression by a broad range of diverse functions.

## 2. CAFs Contribution to Tumor Progression

### 2.1. Desmoplasia

The chronic tissue repair response results in the excessive growth of fibrous or connective tissue around an invasive tumor [10]. This process of desmoplasia is achieved through enhanced ECM production and acquired remodeling ability of CAFs [6,10]. As such, desmoplasia causes the tumor to be hard and stiff, leading to a compromised tumor vasculature with blood vessel collapse and hypoxia, thereby promoting formation aggressive tumor clones and inhibition of drug penetration and uptake [7,11]. Furthermore, CAFs and their generated ECM serve as a physical barrier to tumor infiltration by immune cells, and the stiffness of the ECM has shown to enhance cancer cell invasion [6].

### 2.2. Secretion of Pro-Tumorigenic Modulators

The secretome of CAFs consists of several growth factors and cytokines [8], including connective tissue growth factor (CTGF), epidermal growth factor (EGF), insulin-like growth factor (IGF), hepatocyte growth factor (HGF), basic FGF (bFGF), nerve growth factor (NGF), and IL-6, which promote cancer cell survival as well as its proliferative and invasive behavior. Matrix metalloproteinases (MMPs) released by CAFs remodel the ECM, thus generating permissive tracks to facilitate motility and enhanced invasiveness of cancer cells, eventually boosting metastasis [6,7]. Through the secretion of stromal cell-derived factor 1 (SDF1), next to VEGF-A, CAFs are also able to recruit endothelial progenitor cells into the tumor, thereby stimulating tumor angiogenesis [1,12].

### 2.3. Generation of a Protumorigenic and Immunosuppressive TME

CAFs are generally considered to promote a protumorigenic and immunosuppressive TME [4,6,7,8,13] directly by releasing a plethora of immunosuppressive as well as immunomodulating and stimulating cytokines (e.g., IL-6, IL-4, IL-8, IL-10, IL-11, IL-17A, TNF, TGF-β, and HGF) and chemokines (e.g., CCL2, CCL5, CCL7, CXCL7, CXCL9, CXCL10, CXCL12, and SDF1), inflammatory factors (e.g., prostaglandin E2 (PGE2)) and immune-modulatory molecules (e.g., human leukocyte antigen (HLA)-G) that can retain suppressive immune subsets and impede proper function of cytotoxic lymphocytes, or indirectly via the remodeling of the ECM, forming a physical barrier for immune cell entry [4,13]. In addition to CAF-derived soluble factors CXCL12, IL-6, and TGF-β, CAFs also inhibit antitumor cytotoxicity of CD8^+^ T cells through the acquisition of immune checkpoint molecules, such as programmed cell death ligand 1 (PD-L1), PD-L2, and Fas ligand in several tumor types [13,14]. NK cell cytotoxic activity is mainly counteracted by PGE2 and indoleamine 2,3-dioxygenase (IDO) by reducing cytotoxic molecules (granzyme B and perforin) and cytokine release [13]. In summary, CAFs possess the ability to manipulate the immune system, both by excluding and opposing T and NK cell functions via tumor cell downregulation of antigen presentation, elevated expression of surface inhibitory molecules, and secretion of immunosuppressive factors, as well as by maintaining an aberrant inflammatory protumorigenic environment [8,13]. However, CAFs represent a heterogenous population within the TME, with different CAF subsets exerting distinct functions in tumors [4]. This heterogeneity is illustrated by the presence of both myofibroblastic (i.e., myCAF) and non-myofibroblastic (i.e., iCAF) CAF subpopulations, with associated ECM signature and inflammatory phenotype, respectively, in different cancer types (reviewed in detail) [15].

### 2.4. Mediation of Drug Resistance

CAFs have emerged as key players in mediating drug resistance, and the potential mechanisms to do this are diverse [6,8]. First, CAFs and their generated ECM can function as a physical barrier and thus prevent efficient drug delivery [8]. Moreover, as drug resistance develops over a long period of time, the physical barrier and interaction with the ECM may protect cancer cells from apoptosis while they acquire the mutations needed for cell-intrinsic resistance to therapy [8,16]. Furthermore, CAFs may inhibit the uptake of anticancer drugs as a result of increased intratumoral interstitial fluid pressure, generated by ECM components [6]. Second, the TME might aid cancer cell survival and induce epithelial–mesenchymal transition (EMT) through the enhanced adhesion of cancer cells to the ECM, leading to therapeutic resistance, often described as cell adhesion-mediated drug resistance (CAM-DR) [6,8,17]. As they have been identified as potential drug resistance mediators, CAF-secreted soluble factors TGF-β, IL-6, and HGF may also contribute to therapeutic resistance by binding to cancer cell receptors causing transcriptional changes and by activating non-transcriptional mechanisms including degradation or redistribution of activators of apoptosis (e.g., proapoptotic Bcl-2 member Bim [18] and c-Fas-associated death domain-like IL-1-converting enzyme-like inhibitory protein-long (c-FLIPL)) [19], and increased stability of suppressors of apoptosis and cell cycle regulators (e.g., p27Kip1-associated cell cycle arrest [20]) [6,8]. In addition, upon exposure to conventional chemotherapies, radiotherapy, and targeted agents, CAFs can gain new features with altered functionality and secretion of proteins (e.g., WNT16B [21]) and cytokines (e.g., IL-17A [22]) that drive tumorigenesis, which can ultimately facilitate the development of therapy resistance and contribute to a more aggressive cancer phenotype [7,8].

As CAFs aid tumor development and contribute to therapy resistance [3], CAFs and cancer cells are increasingly viewed as partners in crime. However, the contribution of CAFs is complex and dynamic, with the involvement of various other players of the TME. Given the profound evidence of a generally protumorigenic function of CAFs, there is a considerable heterogeneity and plasticity between different tumor types and divergent context-dependent CAF phenotypes within distinct tumors [6,7,8,23]. Therefore, in this review, we discuss two solid tumor types, i.e., lung cancer and pancreatic cancer, with clear evidence of CAFs contributing to therapy resistance and marked by poor long-term prognosis. We discuss how CAFs contribute to tumor progression, affect their response to clinical therapy and prognosis, and we provide an overview of novel therapies involving CAF-targeting agents for both cancer types.

## 3. Lung Cancer

Lung cancer remains a devastating disease and the leading cause of cancer-related death worldwide [24], independent of sex [25]. Approximately 84.3% of all lung malignancies are classified as non-small cell lung cancer (NSCLC), for which the predicted 5-year relative survival rate for newly diagnosed cases is 21.0% [25] compared to only 7.0% for small cell lung cancer (SCLC) [26].

Important advancements in the treatment of NSCLC have been achieved over the past two decades due to the introduction of small molecule tyrosine kinase inhibitors and immunotherapy [27]. This is in contrast to SCLC, where immunotherapy shows only minor benefit [28]. However, oncogenic driver mutations are only present in 15–20% of NSCLC patients and PD-L1 tumor proportion score (TPS) ≥ 50% in 28% of advanced NSCLC, thereby limiting the use of these novel treatment strategies [29,30]. As such, surgery, cytotoxic chemotherapy, and radiotherapy remain the backbone for the treatment of both NSCLC [24,31] and SCLC [32]. Nonetheless, the overall cure for NSCLC and SCLC remains low, particularly in metastatic disease [27,28], as the majority of patients have already progressed to a more advanced stage at diagnosis, and due to the occurrence of therapeutic resistance, especially to chemotherapy and targeted therapies [33]. Therefore, further molecular characterization of the tumor landscape is needed, in order to investigate resistance mechanisms and develop novel therapeutic strategies [10].

The TME in lung cancer has been recognized to play a central role in the initiation and progression of primary de novo lung cancer [10], as well as in contributing to therapeutic resistance [34]. Lung tumors harbor distinct subsets of CAFs, each expressing a unique repertoire of collagens and other ECM molecules, demonstrating phenotypic diversity in activated pathways, such as EMT and angiogenesis, that may promote invasiveness and metastasis [35]. Next to resistance resulting directly from CAFs, CAFs are able to promote resistance indirectly by modulating the response to chemotherapy, radiotherapy, targeted therapies, and immunotherapy through paracrine signaling to cancer and immune cells and mutual metabolic reprogramming (illustrated in Figure 1) [36].

### 3.1. Resistance to Chemotherapy

Despite the introduction of numerous new treatment modalities, conventional platinum-based chemotherapy remains a pivotal pillar in the treatment of NSCLC, particularly in advanced stages [24,31]. CAFs are able to influence the response of lung cancer cells to chemotherapy through several signaling pathways and through their phenotypic diversity.

CAFs can modulate the response to cisplatin-based chemotherapy through the activation of the Gas6/AXL, IL-11/IL-11R/STAT3, SDF-1/CXCR4/NF-κB/Bcl-xL and SRGN/CD44 signaling pathways. CAF-derived growth arrest specific 6 (Gas6) protein, which is a natural ligand of tumor-associated macrophage (TAM) receptors and which has high affinity to the receptor tyrosine kinase AXL, increases during cisplatin-based chemotherapy. Tumoral AXL activation, which is linked to EMT and promoting cell survival, resistance, invasion, and metastasis in several cancers [37], results in proliferation and migration of lung cancer cells. Moreover, stromal Gas6 and tumoral AXL expression in NSCLC patient samples was associated with a worse 5-year disease-free survival rate [38]. CAFs treated with cisplatin also facilitate chemoresistance of lung adenocarcinoma by releasing IL-11, which results in the upregulation of antiapoptotic protein Bcl-2, thereby decreasing cisplatin-induced apoptosis [39]. By the release of SDF-1, CAFs are capable of promoting cell proliferation and drug resistance of lung cancer cells to cisplatin through upregulation of CXCR4, NF-κB, and Bcl-xL [40]. Furthermore, CAF-derived chondroitin sulfate proteoglycan serglycin (SRGN), a CD44 tumor cell-interacting factor, induces cancer cell stemness via Nanog induction accompanied with increased chemoresistance to cisplatin. In addition, increased expression of SRGN predicted poor prognosis NSCLC patients [41].

Other pathways that are involved in CAF-induced resistance of lung cancer cells to chemotherapy include the ANXA3/JNK, IGF2/AKT/Sox2/PG-P, and PI3K/Akt/GRP78 signaling pathways. CAFs increase Annexin A3 (ANXA3) expression in cancer cells, a Ca^2+^-dependent phospholipid-binding protein with an important role in tumor growth and progression [42]. This leads to the activation of c-jun N-terminal kinase (JNK) and survival, which inhibits cisplatin-induced apoptosis [42]. CAFs also produce IGF2 that binds to the IGF1 receptor in lung cancer cells and activate the AKT/Sox2 pathway, leading to enhanced expression of P glycoprotein (PG-P), a member of the ATP-binding cassette transporters, that is capable of pumping out the chemotherapeutic drug [43]. CAF-derived HGF inhibits paclitaxel-induced apoptosis of lung cancer cells by upregulating the PI3K/Akt pathway and glucose-regulated protein 78 (GRP78), a protective chaperone protein that supports cell survival and is associated with the development of chemoresistance [44].

Regarding the phenotypic diversity of CAFs, a subset with high CD10 and GPR77 expression correlated with chemoresistance and poor survival in breast and lung cancer patients [45]. Consistent with these findings, co-culture of CD10^+^GPR77^+^ CAFs with cancer cells dramatically enhanced survival upon treatment with cisplatin or docetaxel, by providing a constant source of paracrine IL-6 and IL-8 that stimulates the survival of cancer stem cells in the TME, ultimately inducing chemoresistance. Moreover, CD10^+^GPR77^+^ CAFs themselves were resistant to cisplatin and docetaxel by expressing numerous ATP binding cassette transporters [45].

Suppressing alpha-smooth muscle actin (α-SMA) myofibroblastic CAFs—which share the phenotypic traits of fibroblasts and smooth muscle cells, secrete ECM, and generate mechanical tension within tissue through cell contraction [46]—through the inhibition of plasminogen activator inhibitor-1 (PAI-1) limits resistance to cisplatin in lung cancer [47]. CAFs in tissue sections from lung carcinomas incubated with or without cisplatin turned out to be much less sensitive to cisplatin when compared to isolated CAFs, indicating that the TME also attenuates the sensitivity of lung cancer CAFs to cisplatin [48].

CAF-derived IL-6 is also found to be involved in maintenance of a paracrine loop in the interaction between fibroblasts and NSCLC cells. Treatment with cisplatin increases TGF-β production by NSCLC cells, which in turn activates CAFs, resulting in an increase of IL-6 and ultimately contributes to enhancing TGF-β-induced EMT in NSCLC. Furthermore, in human samples, stromal IL-6 expression was significantly correlated with the EMT status of cancer cells and with prognosis in patients, thus indicating that stromal IL-6 secreted from CAFs enhanced EMT signaling and induced resistance against chemotherapy or chemoradiotherapy [49]. In addition, CAF-derived IL-6 enhances the metastatic potential of lung cancer cells by activation of the JAK2/STAT3 signaling pathway [50,51], which in turn also mediates tumor angiogenesis through the upregulation of VEGF and bFGF [52].

In SCLC, the ECM enhances tumorigenicity and confers resistance to chemotherapeutic agents, such as doxorubicin, cisplatin, etoposide, and cyclophosphamide, as a result of β1 integrin-stimulated tyrosine phosphatidyl inositol 3-kinase (PI3-kinase) activation, overriding cell cycle arrest and chemotherapy-induced apoptosis, and allowing SCLC cells to survive with persistent DNA damage [53,54].

### 3.2. Resistance to Radiotherapy

For patients with comorbidities or those that are inoperable and present peripherally located stage I NSCLC, radiotherapy is the preferred treatment. Furthermore, for unresectable locally advanced stage IIIA and IIIB and stage IV NSCLC, radiotherapy is indicated with or without systemic chemotherapy or targeted treatment [24,31].

In general, CAFs are naturally resistant to fractionated radiotherapy, as radiation only causes cytotoxic but not cytolytic effects on CAFs [55]. Irradiated CAFs prior to implantation in mice lose their protumorigenic potential in vivo, as the enhanced tumorigenic effect observed in tumors co-implanted with control CAFs was abrogated. However, CAFs survive single high dose and fractionated radiation doses [56] and promote irradiated lung cancer cell recovery and tumor relapse after radiotherapy by CAF-derived IGF2-induced autophagy via mTOR suppression [57]. Consistent with these findings, CAFs isolated from freshly resected human lung tumors exposed to ablative doses of ionizing radiation show premature cellular senescence and reduced proliferative, migrative, and invasive capacity [58]. In addition, their secretory profile was transformed: the angiogenic factors SDF-1, angiopoietin, and thrombospondin-2 were downregulated in contrast to bFGF that was upregulated. Expression levels of HGF, IL-6, IL-8, IL-1β, and TNF-α remained unchanged. Even though CAFs had a transformed secretory profile, this did not affect the proliferative or migratory capacity of human lung tumor cells [59]. Nonetheless, prematurely induced senescence of human lung fibroblasts after exposure to ionizing radiation enhanced the growth of malignant lung epithelial cells in vitro and in vivo and may contribute to radiotherapy resistance and subsequent lung cancer progression [60]. Interestingly, soluble factors such as TGF-β produced by irradiated fibroblasts promoted murine lung adenocarcinoma cell migration and potential metastatic escape. However, in case of simultaneous irradiation of fibroblasts and carcinoma cells, carcinoma cell migration was repressed [61].

### 3.3. Resistance to Targeted Therapy

Targeted therapies against driver mutations and gene fusions provided significant therapeutic advances for a subset of NSCLC patients [62]. However, the clinical efficacy of such anticancer targeted therapies is strongly limited by the rapid development of acquired drug resistance in the majority of patients [62,63]. The TME has recently emerged as an important player in sustaining resistance to targeted therapies by activation of signals in a paracrine manner, able to compensate the drug-inhibited pathways in tumor cells [63].

Crosstalk between NSCLC cells and CAFs reduces sensitivity to targeted therapies, as CAFs show high expression levels of IL-6 and oncostatin-M (OSM), leading to paracrine JAK1/STAT3 activation in NSCLC cells, thus resulting in a switch to the EMT phenotype and protection from targeted drug-induced apoptosis. Moreover, OSM-receptor (OSMR) gene expression in surgically resected samples appeared to predict worse prognosis for patients with NSCLC, and selective inhibition of JAK1/STAT3 activation and OSMR expression with JAK1 inhibitor filgotinib reversed resistance to targeted drugs [62]. Previous findings also reported the role of CAF-mediated resistance of lung cancer cells to EGFR tyrosine kinase inhibitors (TKI) erlotinib [64] and gefitinib [65] through the induction of an EMT phenotype, suggesting that a specific phenotype of podoplanin- [66] and CD200-expressing [67] CAFs is responsible for the constituting resistance to EGFR TKIs. Despite the presence of podoplanin-expressing CAFs and a relatively poor response to EGFR TKIs among patients with lung adenocarcinoma harboring EGFR-activating mutations, the molecular mechanism remains unclear [66].

Moreover, treatment of lung cancer cells with TKIs targeting *MET* (proto-oncogene) or *EGFR* gene alterations can instruct the TME to sustain an adaptive resistance to targeted therapies by increasing lactate release. This leads to an increased HGF production by CAFs, which in turn activates MET in cancer cells, hence the diminished inhibitory effect of TKIs. Consistently, stromal HGF and tumor cell lactate transporter MCT4 were increased in NSCLC patients who progressed upon EGFR TKI therapy with erlotinib or gefitinib [63]. These results confirmed previous findings that c-MET-amplified tumor cells become dependent on HGF for survival upon pharmacologic MET inhibition [68], as well as for lung cancer cells harboring EGFR mutations upon treatment with EGFR TKI gefitinib [65,69].

### 3.4. Resistance to Immunotherapy

Immune-checkpoint inhibitor treatment, either as monotherapy or combination therapy depending on PD-L1 expression, has been established as the standard of care for patients with locally advanced/metastatic NSCLC without actionable oncogenic driver [24].

CAFs are able to modulate immune responses in the TME of lung cancer, regardless of immunotherapy [10]. CAFs derived from human NSCLC were found functionally and phenotypically heterogeneous and showed a constitutive upregulation of PD-L1 and PD-L2 resulting from autocrine interferon gamma (IFNγ), potentially enhancing or suppressing the activation of T cells. Furthermore, production of several cytokines and chemokines, such as IFNγ and TGF-β1, was demonstrated in these CAFs [70]. A more recent study revealed that CAFs directly contribute to the suppression of antitumor T cell responses by cross-presenting antigens complexed with major histocompatibility complex (MHC) I to antigen-specific CD8^+^ T cells, leading to antigen-specific upregulation of Fas/FasL and PD-1/PD-L2 on T cells and CAFs, respectively, which ultimately results in elimination of tumor-specific T cells and enhanced tumor viability [71].

Not surprisingly, recent studies have highlighted a major role for CAFs in promoting immunotherapy resistance by, at least in part, excluding T cells from tumor mass, which then accumulate at the tumor margin. As such, CAF-rich tumors are clinically aggressive and respond poorly to immunotherapy, as the success of most immunotherapies is dependent on CD8^+^ T cell-infiltrating tumors [72]. Consistently, in samples from patients with NSCLC who did not respond to immunotherapies, two subsets of immunosuppressive CAFs were enriched at time of diagnosis, of which one was able to increase the expression of PD-1 and CTLA-4 at the surface of FOXP3^+^ regulatory T cells (Tregs), which are critical in maintaining immune tolerance and homeostasis of the immune system [73]. Indeed, Tregs coexisting with CAFs are correlated with a poor outcome in NSCLC [74]. As such, the increased CTLA-4 expression by Tregs could explain the additive effect of antibodies blocking CTLA-4 in combination with anti-PD-1 checkpoint inhibitors [73], as this combination has recently been shown to improve overall survival, as compared with chemotherapy, in the first-line setting of patients with metastatic NSCLC [75].

## 4. Pancreatic Cancer

Pancreatic ductal adenocarcinoma (PDAC) is among the deadliest human malignancies. Despite the efforts that have been made, the overall prognosis remains extremely poor, with a 5-year overall survival that still stands below the bar of 10% [76]. Unfortunately, pancreatic cancer is even expected to become the second leading cause of cancer related deaths of the western world within the next decade, due to increasing age and risk factors (e.g., smoking, obesity, and diabetes) [77,78]. These devastating numbers reflect the resistance that is created by the tumor and its environment. Therefore, medical treatment of PDAC remains an ongoing challenge.

PDAC has a unique TME characterized by a dense stromal compartment that can even make up 90% of the whole tumor bulk [79]. This stromal shield is composed of a variety of cell types that are embedded in a dense ECM. CAFs play a central role in the formation of this fibrotic shield and mostly originate from PSCs, a resident stromal cell population of the pancreas [80]. By synthesis and secretion of various ECM components (e.g., collagen, laminin, and fibronectin), CAFs orchestrate the formation of the fibrotic tissue. Due to their presence within the PDAC tumor, along with their secreted factors, they are believed to be the major confounding factor involved in mediating therapeutic resistance (illustrated in Figure 2).

Currently, surgery is the sole curative treatment option for PDAC patients. Unfortunately, most patients present with locally advanced disease or distant metastasis at the time of diagnosis, precluding complete surgical resection of the tumor. Conventional chemotherapy treatment constitutes the standard of care for advanced or metastatic disease. Gemcitabine became the reference chemotherapeutic regimen for advanced PDAC and results in a median overall survival of 6.8 months, or 8.5 months when combined with nab-paclitaxel [81]. Recently, the use of FOLFIRINOX, a combination chemotherapy (oxaliplatin, irinotecan, fluorouracil, and leucovorin) increased the overall survival in these patients to a modest 11.1 months [82]. However, this intensified chemotherapeutic combination is only suitable for patients with a good performance status. Unfortunately, the failure of translating clinical response to chemotherapy into significant survival benefit could be attributed to poor penetration of the administered drug into the tumor and subsequent development of chemoresistance, in which CAFs play a pivotal role [83].

### 4.1. Resistance to Chemotherapy

#### 4.1.1. Extrinsic Resistance

Various mechanisms have been unraveled by which CAFs can mediate chemoresistance in pancreatic cancer. At first, the physical barrier that is established by the stromal shield limits the delivery of administered drugs. Indeed, the dense stromal compartment creates high interstitial pressure primarily generated by hyaluronan [84], a polysaccharide overexpressed by activated CAFs in PDAC [85]. This causes vascular collapse and hypoperfusion, which alters the intratumoral physiology, hence limiting drug delivery. Moreover, increased hyaluronan expression has been correlated with poor prognosis in resected PDAC patients [86]. Therefore, enzymatic degradation of hyaluronan is an emerging strategy for the treatment of PDAC, which is discussed in the section CAF targeted treatment.

A second mechanism of chemoresistance, regulated by CAFs in PDAC, is their ability to alter the chemotherapeutic activity and efficacy by changing gemcitabine metabolism. To exert its antitumor effects, gemcitabine must first be metabolized within the tumor cell itself. Therefore, this hydrophilic molecule needs to be transported through the hydrophobic cell membrane via transmembrane nucleoside transporters, i.e., human equilibrative nucleoside transporter-1 (hENT1) and human concentrative nucleoside transporter-3 (hCNT3). Upon cytoplasmic entry, gemcitabine is phosphorylated by intracellular deoxycytidine kinase to perform its role as a deoxycytidine nucleoside analogue. This nucleoside analogue will then be incorporated into the replicated DNA strand, leading to chain termination, DNA damage, and apoptosis of the cancer cell [83]. However, CAFs are able to alter this crucial gemcitabine metabolism via expression of an arsenal of molecules. First, cysteine-rich angiogenic inducer 61 (CYR61) is a negative regulator of two important nucleoside transporters, i.e., hENT1 and hCNT3. It has been demonstrated that activated CAFs in the pancreatic tumor are a source of CYR61 [87]. Second, activated CAFs are also known to overexpress TGF-β. It has been demonstrated that TGF-β induces CYR61 expression, resulting in inhibition of hENT1 and hCNT3, hence impairing cellular uptake of gemcitabine by the tumor cells [87]. This highlights the potential of targeting CYR61 or TGF-β, to reverse the inhibited cellular uptake of gemcitabine, in order to improve its efficacy in PDAC. Last, CAFs are also able to release deoxycytidine into the TME. Deoxycytidine does not hinder the cellular uptake of gemcitabine, but rather competes downstream for the deoxycytidine kinase within the tumor cell. This results in a limited conversion to the active gemcitabine-triphosphate, thus diminishing its antitumor effect [88]. However, the lack of deoxycytidine secretion is not a property of all fibroblasts, highlighting the heterogeneity of fibroblast subsets, in which deoxycytidine expression might be a phenotypic subtype.

#### 4.1.2. Intrinsic Resistance

The mechanisms described above involve extrinsic mechanisms of CAFs on chemoresistance in PDAC. However, intrinsic mechanisms of chemoresistance of tumor cells itself can also be altered directly or indirectly by CAFs, again helping cancer cells to survive under the influence of chemotherapy. Intrinsic mechanisms such as resistance to apoptosis and enhanced survival capacity of the cancer cells are mediated through various signaling pathways including RAS-RAF-MEK-ERK and PI3K-AKT. This crosstalk between cancer cells and CAFs is established via the expression of a broad range of molecules (e.g., laminin, fibronectin, IGF, SDF-1α, and perlecan), activating various subsequent signaling pathways that promote cancer cell survival [89,90,91,92,93,94]. Different ECM proteins, including laminin and fibronectin, are associated with CAM-DR in different cancer types [95], including pancreatic cancer [96]. Laminin, overexpressed by CAFs, induces phosphorylation of the non-receptor tyrosine kinase known as focal adhesion kinase (FAK), leading to cancer cell survival via the PI-3K-Akt pathway [90]. On the other hand, CAF-secreted fibronectin promotes high ERK1/2 activity in tumor cells, thereby protecting them from gemcitabine-induced cytotoxicity [90]. These findings imply combined use of gemcitabine and laminin or fibronectin inhibitors to counteract chemoresistance in PDAC. Furthermore, stromal-derived IGFs blunt the response to chemotherapy via activation of the insulin/IGF1R survival pathway within the cancer cells [91]. SDF-1, also named CXCL12, is a mediator of tumor–stromal interactions in different cancer types, including lung and pancreatic cancer, via the CXCR4/CXCL12 axis [97]. In PDAC, SDF-1α secretion by surrounding CAFs promotes chemoresistance mediated through paracrine SDF-1α/CXCR4 signaling and subsequently the IL-6 autocrine loop in pancreatic cancer cells. Upregulation of IL-6 in tumor cells, induced by SDF-1α, is due to downstream activation of FAK-AKT and ERK1/2 signaling pathways, thereby supporting cell survival and chemoresistance [92]. Lastl it has been demonstrated that stromal derived perlecan (or heparan sulfate proteoglycan 2, HSPG2) impairs the chemotherapy response in PDAC. In vivo experiments showed that perlecan depletion combined with gemcitabine prolonged mouse survival [94], supporting the potential of anti-stromal treatment for pancreatic cancer to enhance chemotherapy treatment.

#### 4.1.3. Acquired Resistance

It has been demonstrated that chemotherapy is able to induce stromal responses that facilitate tumor progression. This mechanism of acquired resistance, mediated via different pathways, provides a rationale for the failure of chemotherapy following long-term treatment in clinical practice. In vitro experiments unraveled the molecular changes as well as the functional consequences associated with gemcitabine treatment of PDAC CAFs and confirmed that chemotherapy-treated CAFs are more tumor-supportive compared with their untreated counterpart [98]. This was mediated through upregulation of various inflammatory mediators, similar to the senescence-associated secretory phenotype (SASP). Induction of SASP increased the tumor cell viability, migration, and invasion, thus opposing the effects of chemotherapy treatment. Additionally, in vivo experiments showed that inhibition of upstream MAPK in CAFs resulted in attenuation of their tumor-supportive role [98]. This raises the possibility of SASP induction, via its upstream MAPK signaling pathway, as a potential therapeutic target to improve the efficacy of chemotherapy treatment.

Furthermore, gemcitabine treatment can also induce CAFs to secrete exosomes containing different tumor promoting molecules [99]. Studies have shown the production and release of exosomal miRNA-106b and Snai1, which lead to increased survival and enhanced proliferation in recipient cancer cells [99,100]. Additionally, miRNA-451a is highly present in the CAF exosome profile in PDAC [101]. This miRNA has been described to regulate the drug transporter protein P-glycoprotein [102], potentially promoting resistance to chemotherapy that utilize this transporter. However, this has not yet been confirmed in PDAC. Additionally, miR-451 contributes to promoted cell viability in vivo and in vitro and its elevated expression correlated with poor prognosis, including increased tumor invasion and metastasis in PDAC patients [103]. To conclude, the release of exosomes by CAFs offers an additional target to reduce resistance to chemotherapeutics.

### 4.2. Resistance to Radiotherapy

Patients with locally advanced PDAC are often treated with additional radiotherapy. Similar to their role in chemoresistance, CAFs have demonstrated their capacity to impede radiation response by direct or paracrine interaction. Remarkably, radiotherapy enhances the desmoplastic reaction in PDAC via an autocrine periostin loop [104]. Next to its chemoprotective characteristics, desmoplasia is also professed as a mediator of radioprotection via β1-integrin signaling and downstream FAK and MAPK-AKT signaling within tumor cells [105,106]. Furthermore, the desmoplastic reaction creates a hypoxic TME. As minimal levels of oxygen are required to sensitize tumor cells for radiation treatment, the present hypoxia leads to radiotherapy resistance [107,108]. Therefore, hypoxic radiosensitizers might be a rational-based combination strategy to surmount radiotherapy resistance.

Furthermore, in vitro experiments showed that irradiation of CAFs leads to an altered secretome, resulting in radioprotection. Indeed, culturing pancreatic cancer cells with conditioned medium from radiotherapy treated CAFs led to an elevated phosphorylation of c-Met, the HGF receptor. HGF, secreted by CAFs, plays a crucial role in invasion, proliferation and metastasis of cancer cells. Therefore, the overactive HGF-c-Met axis, induced by CAFs upon radiotherapy treatment, supports tumor progression [109]. Radiation of CAFs also promotes the secretion of high concentrations of CXCL12 both in vitro and in vivo via the P38 pathway, as such it promotes pancreatic cancer cell migration, invasion, and EMT [110]. Additionally, both EMT as well as cancer stem cells, modulated by stromal TGF-β secretion, are associated with resistance to radiotherapy [111]. This further supports the future for TGF-β as a therapeutic target for PDAC treatment.

## 5. CAF-Targeted Treatment

As CAFs display a broad range of mechanisms to stimulate tumorigenesis and drug resistance [8], combined with their genetic stability and relative abundance among stromal cells [36], targeting these cells is becoming an appealing and emerging therapeutic strategy. Several different approaches have been suggested for CAF-targeted anticancer treatment: (1) targeting the biophysical stromal barrier to increase drug delivery, (2) inhibiting CAF-secreted factors that stimulate tumorigenesis and drug resistance, (3) depleting or blocking the ECM components to induce stromal depletion and reduce adhesion-mediated signaling, (4) targeting the CAFs themselves to disable their downstream effects, (5) forcing CAFs to adopt an inactive quiescent phenotype (through the use of molecules such as all-trans retinoic acid (ATRA) or calcipotriol), and (6) modifying and using CAFs as in situ durable reservoir to deliver anticancer drugs (such as oncolytic adenoviruses, TNF-related apoptosis-inducing ligand (TRAIL), or IFN) [8,112]. Here, we will highlight the results of CAF-directed anticancer therapies for lung and pancreatic cancer that have been investigated in clinical studies and those under current clinical investigation (Table 1).

### 5.1. CAF-Targeted Treatment in Lung Cancer

CAFs are cells with an extensive proteome and secretome [3]. Therefore, drugs that have been developed successfully the past two decades, such as bevacizumab and atezolizumab, target at the same time CAF-derived VEGF-A and PD-L1 expression by CAFs, respectively, in the TME next to tumoral VEGF-A secretion and PD-L1-expression [10]. Furthermore, compounds such as canakinumab—a human anti-IL-1β monoclonal antibody and currently approved to treat uncommon autoimmune disorders, such as systemic juvenile idiopathic arthritis and periodic fever syndromes [113]—can be repurposed to target the secretome of CAFs, and are currently under investigation in a phase III trial as adjuvant therapy for completely resected (R0) NSCLC stages II-IIIA and IIIB (T > 5 cm N2) (NCT03447769) (Table 1).

In addition, several TKIs that simultaneously inhibit VEGF receptor 1, 2, and/or 3, FGF receptor 1, 2 and/or 3 and/or PDGF receptor α and/or β have been developed (e.g., nintedanib, motesanib, and sorafenib) [10]. These agents block the FGF and PDGF receptors on CAFs preventing activation, and the downstream effects of CAF-derived VEGF and FGF. Nintedanib has currently been approved by the European Medicines Agency (EMA) for use in combination with docetaxel for the treatment of patients with advanced NSCLC with disease progression after frontline platinum-based chemotherapy [114]. On the other hand, motesanib [115] and sorafenib [116], next to other similar compounds (reviewed by Altorki et al. [10]) all failed to show any benefit in phase III trials.

Rogaratinib, a FGFR-selective small-molecule kinase inhibitor of FGF receptor 1–4 aiming to prevent CAF activation, demonstrated in a phase I study partial response (PR) in 1 patient, stable disease (SD) in 15 patients, and progressive disease (PD) in the remaining 4 patients of the NSCLC cohort [117]. Infigratinib, a FGF receptor 1–3 inhibitor, also showed promising results with disease control in 18 (50%) out of 36 patients with FGFR1-amplified squamous NSCLC; four patients achieved PR and 14 had SD [118]. Currently, a phase II trial is investigating erdafitinib, a TKI of FGF receptor 1–4, in advanced solid tumors including NSCLC (NCT02699606) (Table 1).

Other specific CAF-targeted treatments using different approaches have been explored.

Sibrotuzumab, a 131I-radiolabeled anti-FAP monoclonal antibody on activated CAFs, was evaluated in a phase I study in patients with advanced or metastatic fibroblast activation protein (FAP) positive cancer, including NSCLC next to colorectal carcinoma. Despite sibrotuzumab was well tolerated and safe with repeat infusions, only two patients (one patient with NSCLC and one patient with colorectal carcinoma) had SD at 12 weeks, the remaining 21 patients showed PD at restaging [119]. In a phase II clinical trial for metastatic colorectal cancer, the anti-FAP monoclonal antibody failed as the minimal requirement for the continuation of the trial was not met [120]. Defactinib, an inhibitor of FAK expressed on CAFs and involved in inducing cell motility, extracellular matrix deposition, survival, and proliferation [121], demonstrated only modest clinical activity in heavily pretreated patients with KRAS mutant NSCLC with 15 (28%) patients that met the 12-week progression free-survival endpoint and with one patient achieving a PR [122].

Targeting the CAF secretome with S-3304, an MMP-inhibitor which most potently inhibits the activities of CAF-derived MMP-2 and MMP-9, in patients with advanced and refractory solid tumors, including NSCLC, showed no complete (CR) or PR, but only SD in seven out of 26 patients, of which none had NSCLC [123]. Furthermore, celecoxib, a selective COX-2 inhibitor and also part of the CAFs proteome [8], showed no significant improvement in PFS in combination with standard chemotherapy vs. placebo and chemotherapy in NSCLC patients with COX-2 overexpression [124]. A phase II study of belagenpumatucel-L, a TGF-β2 antisense gene-modified allogeneic tumor cell vaccine to inhibit cancer cell-derived (but not CAF-derived) TGF-β2, as maintenance therapy in NSCLC failed to meet its survival endpoint [125].

Currently, several histone deacetylase (HDAC) inhibitors, such as entinostat, vorinostat, and mocetinostat, in combination with PD-1 or PD-L1-inhibitors, are under clinical investigation in NSCLC. These agents can alter epigenetic regulation and intracellular signal transduction, including the JAK1/STAT3 pathway, not only in cancer cells, but also in CAFs or precursors of CAFs. As such, HDAC inhibitors potentially can reduce the generation and activation of CAFs and thus eliminate CAF infiltration in the TME [112].

### 5.2. CAF-Targeted Treatment in Pancreatic Cancer

Due to the central role of CAFs in impeding both chemo- and radiotherapy response in PDAC, stromal depletion swiftly emerged as a tempting therapeutic strategy. Preclinical models demonstrated the benefit of CAF ablation through blockade of the hedgehog signaling pathway, resulting in enhanced chemotherapeutic delivery [126]. Unfortunately, the use of different hedgehog inhibitors showed no benefit compared to gemcitabine or FOLFIRINOX monotherapy. Additionally, these clinical trials were halted due to a paradoxical accelerated disease progression as a result of more aggressive tumors [127,128,129,130], suggesting the presence of CAF subtypes carrying a tumor-restraining role, which has been confirmed in preclinical experiments [131]. This highlights the need for precise characterization of CAF heterogeneity in the context of developing effective therapies that target relevant CAF subtypes, while retaining the normal fibroblasts.

Additionally, an era of therapies emerged to target the ECM components with the purpose of reversing the desmoplastic reaction and thereby enhancing chemo- and radiotherapy [132]. High hopes were raised for PEGPH20 (pegylated hyaluronidase), as high hyaluronan is the main ECM component responsible for the elevated interstitial pressure [84] and correlated with poor prognosis in PDAC [86]. Unfortunately, the promising results obtained in a phase I/II clinical trial [133] were not translated in the subsequent phase III HALO 301 study [134]. On the other hand, losartan, an angiotensin receptor blocker, has been shown to reduce expression of TGF-β, HA synthases, and collagen I by CAFs [135] and is currently investigated in two clinical trials (Table 1). Prospective evaluation of losartan combined with FOLFIRINOX resulted in downstaging from locally advanced PDAC to a resectable tumor and prolonged survival [136].

As therapy resistance is also modulated via signals secreted by CAFs, blocking these signals might enhance current therapy options for PDAC patients. The central role of CAF-secreted CXCL12 in mediating both chemo- and radiotherapy resistance raised interest to target the CXCL12/CXCR4 axis (Table 1). Preclinical data on targeting this axis showed increased T cell infiltration, precluding its potential to be utilized in combination with immunotherapy [137]. Furthermore, TGF-β, a major player of the CAF secretome, is involved in therapy resistance via various mechanisms, described in previous sections. Therefore, therapies targeting this molecule were investigated in clinical trials [138]. However, addition of the TGF-β inhibitor galunisertib only led to a slight increase in overall survival versus gemcitabine treatment alone in unresectable pancreatic cancer patients [139].

Not only the broad range of CAF functions, but also additional subtype heterogeneity, introduces a challenge to the field of CAF targeting, which is demonstrated by the discouraging results in clinical trials. Therefore, clinical benefit might require targeting of CAF subtypes or reprogramming of CAFs to a normal fibroblast or an antitumorigenic phenotype. Indeed, more clinical trials targeting the specific FAP^+^ CAF subtype are emerging (Table 1), however results from these trials are not published yet. Moreover, CAF reprogramming is currently investigated with the use of paricalcitol and ATRA (Table 1). Although, it remains important to unravel if different states of CAF activation are represented by different subtypes, mediating various mechanisms of therapy resistance.

## 6. Conclusions and Future Perspectives

There is no doubt that CAFs mediate tumorigenesis and therapy resistance in lung and pancreatic cancer, and the mechanisms in doing so are diverse and ingenious, depending on the type of therapy. Therefore, targeting CAFs to counteract these protumorigenic effects and overcome the resistance to current therapeutic options is an appealing and emerging strategy. Moreover, their relative abundance in the TME and genetic stability [36] make them a valuable and sustainable target, as they are less prone to develop resistance phenotypes as a result of high mutation rates and clonal selection compared to cancer cells [8]. However, despite accumulating preclinical research exploring CAF-targeted cancer therapies [7,10,112], clinical trials with these agents have been disappointing so far (reviewed in detail [112]).

Substantial CAF heterogeneity is present in both lung [35] and PDAC [140,141,142] and is probably underestimated. Indeed, CAFs can originate from several different cell types and multiple mechanisms underlying CAF heterogeneity have been proposed (reviewed in detail) [142,143]. In combination with the lack and complex identification of specific CAF markers, CAF heterogeneity with distinct phenotypes and function provide a possible rationale for the current failure of clinical CAF-targeted treatments. Furthermore, differential effects between preclinical and clinical settings might be explained by the current challenge to construe preclinical models that precisely recapitulate this complex heterogeneity.

Therefore, preclinical in vitro 3D culture technologies, such as organoids, are opening avenues for the development of novel, more physiological human cancer models that incorporate CAFs [144] in contrast to conventional 2D homogenous cell culture experiments. Furthermore, a better understanding and identification of the CAF subpopulations that trigger resistance and the molecular mechanisms behind it, could tell us how CAFs can be effectively and specifically targeted to halt therapy resistance. Due to emerging molecular single cell analyzing strategies, more CAF subtypes will be defined. Digital deconvolution of bulk tumor expression profiling datasets demonstrated the presence of both activated and normal stromal signatures in PDAC [145]. Furthermore, distinct transcriptional profiles are displayed among subgroups of CAFs in lung [35] and PDAC [140,141], and epigenetic reprogramming (e.g., miRNA status and DNA methylation status) also contributes to the complexity of defining different CAF subtypes [142,143]. However, it is not merely about different CAF subtypes; mirroring their activation status is necessary to gain more insight in how resistance is governed by CAFs. Unravelling their molecular composition and epigenetic status might tell us how relevant CAF subtypes can be targeted specifically, while avoiding normal fibroblasts. We foresee that, in the future, therapies must be coupled with thorough molecular characterization of present CAFs in each individual to ensure optimal therapeutic efficacy and effectively overcome resistance.

## Figures and Tables

**Figure 1 cancers-13-00987-f001:**
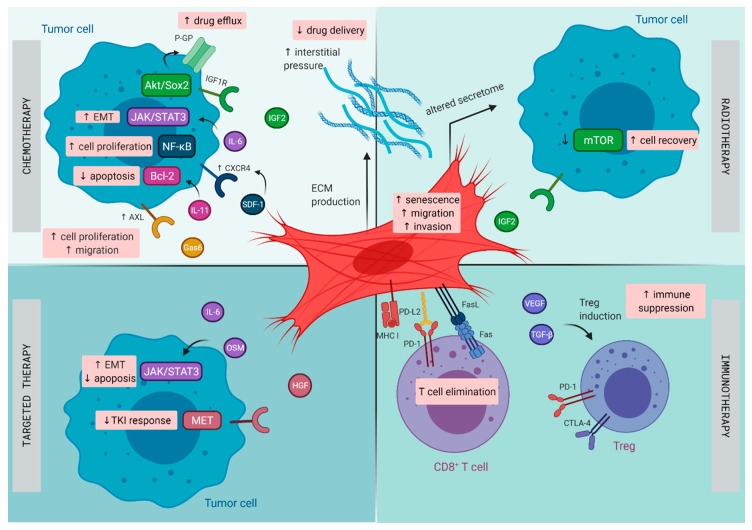
Mechanisms of therapy resistance in lung cancer orchestrated by cancer -associated fibroblasts (CAFs): Schematic illustration on how CAFs diminish the effect of chemotherapy, radiotherapy, targeted therapy, and immunotherapy in lung cancer. CTLA-4, cytotoxic T lymphocyte-associated protein 4; CXCR4, C-X-C chemokine receptor type 4; ECM, extracellular matrix; EMT, epithelial–mesenchymal transition; FasL, Fas ligand; Gas6, growth arrest specific-6 protein; HGF, hepatocyte growth factor; IGF1R, insulin-like growth factor 1 receptor; IGF2, insulin-like growth factor 2; IL, interleukin; MHC I, major histocompatibility complex I; OSM, oncostatin-M; PD-1, programmed cell death 1; PD-L2, programmed cell death ligand 2; P-GP, P-glycoprotein; SDF, stromal cell-derived factor; TGF-β, transforming growth factor-β; TKI, tyrosine kinase inhibitor; Treg, regulatory T cell; VEGF, vascular endothelial growth factor.

**Figure 2 cancers-13-00987-f002:**
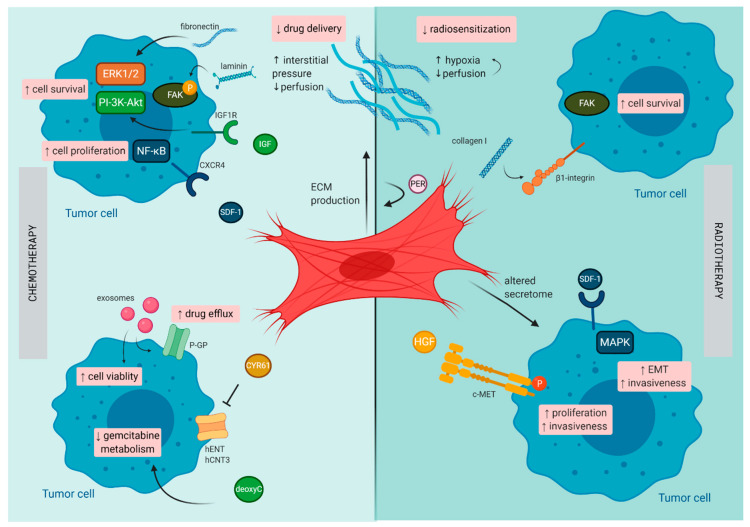
Mechanisms of therapy resistance in pancreatic cancer orchestrated by cancer -associated fibroblasts (CAFs): Schematic illustration on how CAFs diminish the effect of chemotherapy and radiotherapy in pancreatic cancer. c-MET, tyrosine-protein kinase Met CXCR4, C-X-C chemokine receptor type 4; CYR61, cysteine-rich angiogenic inducer 61; deoxyC, deoxycytidine; EMT, epithelial–mesenchymal transition; FAK, focal adhesion kinase; hCNT3, human concentrative nucleoside transporter-3; hENT1, human equilibrative nucleoside transporter-1; HGF, hepatocyte growth factor; IGF, insulin-like growth factor; P, phosphorylated; PER, periostin; P-GP, P-glycoprotein; SDF-1, stromal cell-derived factor-1.

**Table 1 cancers-13-00987-t001:** Active clinical trials targeting CAF in lung cancer and PDAC.

**Lung Cancer**						
**Goal**	**Compound**	**Class**	**ID**	**Title**	**Phase**	**Status**
Reduce ECM and IL-1β	Canakinumab	mAB targeting IL-1β	NCT03447769	Canakinumab as Adjuvant Therapy in Adult Subjects with Stages AJCC/UICC v. 8 II-IIIA and IIIB (T > 5 cm N2) Completely Resected NSCLC	Phase III	Recruiting
Prevent activation of CAFs and block CAF secretome	Erdafitinib	TKI of FGF receptor 1-4	NCT02699606	Erdafitinib, A Pan- FGFR Tyrosine Kinase Inhibitor, In Asian Participants with Advanced NSCLC, Urothelial Cancer, Esophageal Cancer Or Cholangiocarcinoma	Phase II	Active, not recruiting
Prevent generation and activation of CAFs	Entinostat	HDAC inhibitor	NCT02437136	Entinostat with Pembrolizumab in NSCLC with Expansion Cohorts in NSCLC, Melanoma, and Colorectal Cancer	Phase II	Active recruiting
			NCT01928576	Epigenetic Therapy with Azacitidine and Entinostat with Concurrent Nivolumab in Subjects With Metastatic NSCLC	Phase II	Recruiting
Prevent generation and activation of CAFs	Vorinstat	HDAC inhibitor	NCT02638090	Pembrolizumab and Vorinostat in Patients with Immune Therapy Naïve and Immune Therapy Pretreated Stage IV NSCLC	Phase I/II	Active recruiting
Prevent generation and activation of CAFs	Mocetinostat	HDAC inhibitor	NCT02805660	Mocetinostat and Durvalumab in Patients with Advanced Solid Tumors and NSCLC	Phase II	Completed, no results yet
**PDAC**						
**Goal**	**Compound**	**Class**	**ID**	**Title**	**Phase**	**Status**
Reduce ECM and IL-1β	Canakinumab	mAB targeting IL-1β	NCT04581343	A Phase 1B Study of Canakinumab, Spartalizumab, Nab-paclitaxel, and Gemcitabine in Metastatic PC Patients (PanCAN-SR1)	Phase Ib	Recruiting
Reduce ECM and TGF-β	Losartan	Angiotensin II receptor antagonist	NCT03563248	Losartan and Nivolumab in Combination with FOLFIRINOX and SBRT in Localized Pancreatic Cancer	Phase II	Active recruiting
			NCT01821729	Proton w/FOLFIRINOX-Losartan for Pancreatic Cancer	Phase II	Active not recruiting
Normalize CAFs	ATRA	Vitamin A derivative	NCT04241276	Phase IIb Randomized Trial of ATRA in a Novel Drug Combination for Pancreatic Cancer (STARPAC2)	Phase IIb	Not yet recruiting
Normalize CAFs	Paricalcitol	Vitamin D analogue	NCT03520790	Paricalcitol Plus Gemcitabine and Nab-paclitaxel in Metastatic Pancreatic Cancer	Phase I/II	Active, not recruiting
Reduce CAF secretome	Plerixafor	CXCR4 antagonist	NCT04177810	Plerixafor and Cemiplimab in Metastatic Pancreatic Cancer	Phase II	Recruiting
Reduce CAF secretome	GSK2256098	FAK inhibitor	NCT02428270	A Study of GSK2256098 and Trametinib in Advanced Pancreatic Cancer	Phase II	Active, not recruiting
Target FAP^+^ CAFs	BXCL701 (talabostat)	Small molecule inhibitor of FAP and dipeptidyl peptidases	NCT04123574	A Pilot Study of BXCL701 in Patients With Pancreatic Cancer	Early phase I	Recruiting
Target FAP^+^ CAFs	CAR-T targeting nectin4/FAP	CAR-T cell	NCT03932565	Interventional Therapy Sequential With the Fourth-generation CAR-T Targeting Nectin4/FAP for Malignant Solid Tumors	Phase I	Recruiting

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
