# Peer review of "Cancer-Associated Fibroblasts as a Common Orchestrator of Therapy Resistance in Lung and Pancreatic Cancer"

_cancers, 2021, doi:10.3390/cancers13050987_

Round 1
Reviewer 1 Report
In this review, Andreas Domen et al described the role of Cancer-associated fibroblasts (CAFs) in the resistance to current therapeutic options in lung and pancreatic cancer. In addition, authors discuss the efficacy of the novel therapies involving CAF- targeting agents. The manuscript covers several aspects of the subject and is well organized. The paper is an update on the topic.
However, the introduction as well as the paragraphs concerning lung cancers are too long and need to be rewritten.
For the clarity of the paper, the authors can use short sentences.
The best characterization of DNA repair proteins and pathways can help the understanding of the interaction between cancer cells and their microenvironment such as fibroblasts essentially in lung cancers. The greatest absent in this paper is the genomic instabilities: microsatellite instability and chromosomal instability. The role of genomic instability and clinical outcomes should be more described regarding the new therapeutic strategy.
The authors need to pay attention to some abbreviations.
Author Response
As correctly pointed out by the reviewer, the introduction as well as the paragraphs concerning lung cancer are too long. Therefore, we have shortened our introduction by adding an additional separate section ‘CAFs contribution to tumor progression’ that summarizes the function of CAFs. We also shortened the introductory paragraph of lung cancer and deleted repetitions.
The reviewer indicates the absence of genomic instability (i.e., microsatellite instability and chromosomal instability). However, recent literature states that genetic aberrations are rare in CAFs, and mainly focused on colorectal cancer [1,2]. On the other hand, epigenetic alterations, upon interaction with neighboring tumor cells can alter the CAF phenotype. Although not much is known on how they interfere with CAF-mediated drug resistance. Therefore, we briefly included this topic within the section ‘conclusions and future perspectives’.
- Pereira, B.A.; Vennin, C.; Papanicolaou, M.; Chambers, C.R.; Herrmann, D.; Morton, J.P.; Cox, T.R.; Timpson, P. CAF Subpopulations: A New Reservoir of Stromal Targets in Pancreatic Cancer. Trends Cancer 2019, 5, 724-741, doi:10.1016/j.trecan.2019.09.010.
- Du, H.; Che, G. Genetic alterations and epigenetic alterations of cancer-associated fibroblasts. Oncol Lett 2017, 13, 3-12, doi:10.3892/ol.2016.5451.
Reviewer 2 Report
The manuscript by Domen et al. well describes the role of CAF in various aspects of cancer.
The only criticism I want to raise is that the Introduction section is too long and repeats some concepts several times, such as the CAF function. It would be better to make a short introduction focused on the state of the art and then make a paragraph on the CAF function on tumors.
Also in Table 1, it may be better to delete the title of the article and instead insert the number of the bibliographic reference.
Author Response
As pointed out by the reviewer, the introduction section is too long and repeats some concepts several times, such as the CAF function, and suggested to make a short introduction focused on the state-of-the-art and create a paragraph on the CAF function in tumors. Therefore, we have shortened our introduction by adding an additional separate section ‘CAFs contribution to tumor progression’ that summarizes the function of CAFs on tumors. We also shortened the introductory paragraph of lung cancer and deleted repetitions on CAF function.
The reviewer also suggested to delete the title of the article and instead insert the number of the bibliographic reference. However, this table summarizes ongoing clinical trials, hence including articles is not applicable yet.
Reviewer 3 Report
The article entitled as (Cancer-associated fibroblasts as a common orchestrator of therapy resistance in lung and pancreatic cancer ) discussed the role of CAFs in promoting cancer aggressiveness. The authors did a great job covering most of the topics. However, the following points need to be addressed :
1- The authors mentioned general statements without supporting that by examples, such as the following
- In line 42, examples of the innate and adaptive immune system
- In line 59, cytokines and chemokines
- In line 71, growth factors
- In line 102, define the abbreviation FAS
- In line 107, surface inhibitory molecules, secretion 107 of immunosuppressive factors
- In line 128, what are the examples of cell apoptotic activators and suppressors
- In line 131, what kind of protein will be secreted
- In line 136, what are the other players?
- In line 149, the statistic is outdated; please consider adding new statistic ( 2020 or 2021)
- 2- The authors are asked to add once section regarding stroma depletions drugs in both NSCLC and PDAC.
- 3- KRAS is one of the major factors that enhance the tumor aggressiveness in both PDAC and NSCLC, the authors are asked to discuss the KRAS role in these tumors , and add the latest trials to target KRAS in both PDAC and NSCLC
Author Response
The reviewer requested to add examples on general statements made in the manuscript. Therefore, we added examples to all general statements as suggested. However, ‘FAS’ (line 102) is not an abbreviation, but has been changed to ‘Fas’ to prevent misinterpretation. Furthermore, ‘surface inhibitory molecules’ and ‘secretion of immunosuppressive factors’ (line 107) is a summary and refers to PD-L1, PD-L2 and Fas ligand (line 102), and to chemokines and cytokines (line 93 to 96) describe above.
The reviewer proposed to add one section regarding stroma depleting drugs in both NSCLC and PDAC. However, the main findings on this topic that are within the scope of drug resistance, were listed per tumor type and ongoing trails are included in table 1. Therefore, adding an extra section seems redundant to our opinion. However, with regard to this interesting topic, we included an extra reference where this subject is reviewed in great detail.
Lastly, the reviewer also suggested to discuss the role of KRAS mutations in these tumor types, and add the latest trials to target KRAS in both PDAC and NSCLC. However, within this manuscript and further literature, to our knowledge, no direct link of KRAS and CAF-mediated drug resistance is described. Therefore, we did not include this section within this manuscript.
Reviewer 4 Report
Domen et al. reviewed the influence of cancer associated fibroblasts on the tumor microenvironment and detail their emerging role in drug resistance. The introduction provides a concise description of the current understanding of CAFs and how they are molecularly and functionally different from normal fibroblasts. The main sections cover the mechanisms of drug resistance and the protumorigenic influence of CAFs specifically in lung and pancreatic cancers. A more detailed rationale could be provided around the decision to focus on these two tumor types – has there been more clinical success in CAF-targeted therapies for these cancers? Finally, CAF-targeted therapies are described including an outline of CAF-related clinical trials. More detail of CAF heterogeneity and plasticity could be useful as it is reportedly impeding the success of CAF-targeted therapeutic strategies in clinical trials. Overall, the authors provide an interesting and comprehensive look at CAFs and their interplay with different therapies, and the figures are illustrated well and informative.
Author Response
The reviewer requested to provide a more detailed rationale around the decision to focus on these two tumor types and whether there is been more clinical success in CAF-targeted therapies for these cancers? We opted for these two tumor types as these have clear evidence of CAFs and as well as poor long-term prognosis. To address this concern, we changed the sentence on line 142 to ‘…with clear evidence of CAFs contributing to therapy resistance and marked by poor long-term prognosis.’ to emphasize the rationale for this decision.
The reviewer also requested more detail on CAF heterogeneity and plasticity as it is reportedly impeding the success of CAF-targeted therapeutic strategies in clinical trials. Therefore, we added the following sentences in Conclusions and future perspectives to emphasize CAF heterogeneity and how preclinical in vitro 3D culture technologies can lead to success of CAF-targeted therapeutic strategies;
- Indeed, CAFs can originate from several different cell types and multiple mechanisms underlying CAF heterogeneity have been proposed (reviewed in detail) [1,2].
- Therefore, preclinical in vitro 3D culture technologies, such as organoids, are opening avenues for the development of novel, more physiological human cancer models that incorporate CAFs [3] in contrast to conventional 2D homogenous cell culture experiments.
References
- Pereira, B.A.; Vennin, C.; Papanicolaou, M.; Chambers, C.R.; Herrmann, D.; Morton, J.P.; Cox, T.R.; Timpson, P. CAF Subpopulations: A New Reservoir of Stromal Targets in Pancreatic Cancer. Trends Cancer 2019, 5, 724-741, doi:10.1016/j.trecan.2019.09.010.
- Du, H.; Che, G. Genetic alterations and epigenetic alterations of cancer-associated fibroblasts. Oncol Lett 2017, 13, 3-12, doi:10.3892/ol.2016.5451.
- Drost, J.; Clevers, H. Organoids in cancer research. Nature Reviews Cancer 2018, 18, 407-418, doi:10.1038/s41568-018-0007-6.